# Imaging-Based Staging of Hepatic Fibrosis in Patients with Hepatitis B: A Dynamic Radiomics Model Based on Gd-EOB-DTPA-Enhanced MRI

**DOI:** 10.3390/biom11020307

**Published:** 2021-02-18

**Authors:** Rencheng Zheng, Chunzi Shi, Chengyan Wang, Nannan Shi, Tian Qiu, Weibo Chen, Yuxin Shi, He Wang

**Affiliations:** 1Institute of Science and Technology for Brain-Inspired Intelligence, Fudan University, Shanghai 200433, China; 18110850012@fudan.edu.cn; 2Key Laboratory of Computational Neuroscience and Brain-Inspired Intelligence (Fudan University), Ministry of Education, Shanghai 200433, China; 3Department of Radiology, Shanghai Public Health Clinical Center, Fudan University, Shanghai 201052, China; SCZ891202@163.com (C.S.); shinannan@shphc.org.cn (N.S.); 16211300007@fudan.edu.cn (T.Q.); 4Human Phenome Institute, Fudan University, Shanghai 200433, China; wangcy@fudan.edu.cn; 5Market Solutions Center, Philips Healthcare, Shanghai 200072, China; Weibo.chen@philips.com

**Keywords:** deep learning, dynamic radiomics analysis, Gd-EOB-DTPA, hepatitis B, hepatic fibrosis, time-domain information

## Abstract

Accurate grading of liver fibrosis can effectively assess the severity of liver disease and help doctors make an appropriate diagnosis. This study aimed to perform the automatic staging of hepatic fibrosis on patients with hepatitis B, who underwent gadolinium ethoxybenzyl diethylenetriamine pentaacetic acid (Gd-EOB-DTPA)-enhanced magnetic resonance imaging with dynamic radiomics analysis. The proposed dynamic radiomics model combined imaging features from multi-phase dynamic contrast-enhanced (DCE) images and time-domain information. Imaging features were extracted from the deep learning-based segmented liver volume, and time-domain features were further explored to analyze the variation in features during contrast enhancement. Model construction and evaluation were based on a 132-case data set. The proposed model achieved remarkable performance in significant fibrosis (fibrosis stage S1 vs. S2–S4; accuracy (ACC) = 0.875, area under the curve (AUC) = 0.867), advanced fibrosis (S1–S2 vs. S3–S4; ACC = 0.825, AUC = 0.874), and cirrhosis (S1–S3 vs. S4; ACC = 0.850, AUC = 0.900) classifications in the test set. It was more dominant compared with the conventional single-phase or multi-phase DCE-based radiomics models, normalized liver enhancement, and some serological indicators. Time-domain features were found to play an important role in the classification models. The dynamic radiomics model can be applied for highly accurate automatic hepatic fibrosis staging.

## 1. Introduction

Hepatic fibrosis is a common pathological process in a variety of chronic liver diseases. It reflects the response to liver damage due to various causes. During hepatic stellate cell proliferation, large amounts of extracellular matrix components are deposited in the extravascular space to cause hepatic fibrosis [1]. Evidence indicates that treatment is necessary for patients with hepatic fibrosis Scheuer–Ludwig ≥S2 [2]. Liver fibrosis can be reversed with antiviral and antifibrotic treatment, even in early cirrhosis [3,4]. Therefore, accurate diagnosis of different grades of liver fibrosis is the prerequisite for evaluating the liver disease status in patients and providing effective and reasonable treatment.

Liver biopsy is the gold standard for diagnosing liver fibrosis; however, approximately 2% of patients experience symptoms such as bleeding and infection during the examination due to the invasiveness of the test. Therefore, this method is not acceptable to all patients, especially those with low-grade fibrosis and relatively stable disease [5]. Sampling errors may occur in biopsy analysis due to the diffuse nature of liver fibrosis [1] and limited material collected, and low repeatability of specimen collection is observed for different operators [6]. As a result, identifying an accurate, non-invasive, safe, simple, reproducible, and long-term follow-up examination approach has been a hot spot in clinical studies.

By definition, radiomics is an approach that extracts quantitative features, including gray-scale patterns, inter-pixel relationships, and shape and spectral properties, from specific regions of interest (ROI) in medical images. Some of these pivotal features can be further applied to generate computational models based on machine learning algorithms, so as to address clinical issues and provide treatment guidance [7]. Radiomics has been adopted in studies assessing various diseases for risk prediction, tumor typing, survival prediction, classification, and staging [8,9,10,11]. In addition, previous radiomics research involved liver fibrosis classification [12,13,14,15,16,17,18].

Gadolinium ethoxybenzyl (EOB) diethylenetriamine pentaacetic acid (Gd-EOB-DTPA; Bayer Health Care Co., Ltd.) is a new hepatocyte-specific magnetic resonance imaging (MRI) contrast agent that can accelerate research and development in liver imaging. The hepatocyte absorbance of Gd-EOB-DTPA is higher than those of conventional agents due to the presence of a lipophilic EOB group in the molecular structure [19]. Moreover, the biochemical properties of Gd-EOB-DTPA provide an overall assessment of tissue perfusion in the arterial phase, while also assessing the specific accumulation in the liver hepatobiliary phase (HBP) after 20 min [20,21]. Studies pointed out that liver-specific Gd-EOB-DTPA-enhanced MRI had great potential to assess liver function and liver fibrosis [22,23,24,25,26,27].

### 1.1. Related Studies

#### 1.1.1. Liver Stiffness Measurement

Amon ultrasound-based techniques, shear wave elastography and acoustic radiation force impulse (ARFI) techniques are commonly used to stage liver fibrosis based on non-invasive tissue stiffness [28]. Both Cassinotto and Friedrich and their collaborators noted that liver stiffness could be detected by ultrasound elastography to assess the extent of fibrosis [29,30]. However, this method is unreliable because of no reproducibility and operator dependence [31]. Magnetic resonance elastography (MRE) is a non-invasive MRI-based technique that can be used to quantitatively evaluate the mechanical properties of body tissues [32]. Many studies demonstrated that hepatic stiffness, measured by MRE, had a strong correlation with the stage of fibrosis according to histology [33,34,35]. The acquisition time of MRE can be less than a minute. In addition, MRE is much less operator-dependent, and has a low rate of technical failure compared with ultrasound-based techniques [32]. However, compared with conventional MRI, MRE requires additional equipment and is more expensive. In addition, the evaluation of liver stiffness-based hepatic fibrosis may be limited by confounders of increased tissue stiffness, which were previously analyzed in detail [36].

#### 1.1.2. Radiomics Analysis

Multiple studies based on radiomics analysis have been proposed for hepatic fibrosis assessment. Duan et al. [12] evaluated hepatic fibrosis in rats based on vessel features, including the textural features of the vessel’s inner wall, by high-resolution computed tomography (CT) using diffraction-enhanced imaging. Zhang et al. [13] compared texture patterns in CT and MRI for hepatic fibrosis staging and found that MRI images had an advantage over CT findings. Kato et al. [14] performed texture analysis on T1-weighted MRI images, with seven texture features extracted using the finite difference method and processed by an artificial neural network program. In a study by House et al. [15], 14 texture features from T2-weighted images were used to investigate the ability of texture analysis to stage liver fibrosis in patients with a range of liver diseases. Cannella et al. [16] investigated the performance of texture analysis in T1-weighed MRI images for hepatic fibrosis evaluation in patients with nonalcoholic fatty liver disease (NAFLD). In studies involving the HBP, Wu et al. [17] assessed the feasibility of texture analysis for hepatic fibrosis staging on T2-weighted, T1-weighted, and Gd-EOB-DTPA-enhanced hepatocyte-phase images in patients with hepatitis C, from which 279 texture features were extracted in a circular ROI for each sequence, including 30 selected for the classification. Park et al. [18] assessed a large data set including patients with 436 pathologically proven liver fibrosis and performed a radiomics analysis in the HBP; the radiomics model significantly outperformed for clinical parameters commonly used for liver fibrosis assessment.

#### 1.1.3. Deep Learning Model

Deep learning has been widely reported for detection, classification, and segmentation of lesions in recent years. In liver imaging, deep learning has also been used for assessing of liver fibrosis. Wang et al. [37] adopted a neural network to extract radiomics features that could provide some high-level features and result in a considerable staging performance for hepatic fibrosis in two-dimensional shear wave elastography (2D-SWE). Yasaka et al. [38] applied a deep convolutional neural network (DCNN) in cropped CT images and found areas under the curves (AUC) of 0.73–0.76 for significant fibrosis, advanced fibrosis, and cirrhosis classification. In another study by Yasaka et al. [39], DCNN in cropped HBP MRI images was used, and AUCs of 0.84–0.85 were obtained for liver fibrosis grading. Recent studies collected larger data sets and achieved higher accuracy [37,40]. Deep learning models are powerful, but highly data dependent and therefore not suitable for small-sample studies, leading to severe overfitting.

### 1.2. Our Contributions

Although many studies were performed for automatic liver fibrosis grading with radiomics analysis or deep learning algorithms, few studies reported using time-domain information, that is, the variation of imaging features in the time series. We proposed a novel dynamic radiomics model combining imaging features from multi-phase DCE images and time-domain features through the contrast enhancement process to assess liver fibrosis. The proposed model outperformed conventional single-phase or multi-phase-based radiomics models, normalized liver enhancement (NLE), as well as some clinical serum parameters.

The main contributions of our liver fibrosis staging pipeline were as follows:Time-domain information was fully used based on time-varying curves and discrepancy of imaging features with the contrast enhancement process, which was found to play a critical role in all stages of classification.ROI extraction was based on whole-liver region segmentation using three-dimensional (3D) U-net and transfer learning, with the post-processing algorithm for interference information excluded. This method of ROI extraction replaced manual selection and delineation, eliminating the influence of region selection on the results of radiomics analysis while being more automated.Feature extraction was performed in liver volume and more valuable information, such as morphological changes in the liver, was considered, which facilitated the classification of cirrhosis.

## 2. Materials and Methods

### 2.1. Dataset

#### 2.1.1. Study Population

This study was approved by the ethics committee of Shanghai Public Health Clinical Center (YJ-2019-S037–02; 05/11/2019). The sample comprised mainly patients with chronic hepatitis B and Child-Pugh score <7. A total of 132 participants were recruited from August 2016 to February 2019. These patients, who underwent liver aspiration biopsy, were divided into four subgroups, including the S1 (n = 30), S2 (n = 28), S3 (n = 32), and S4 (n = 42) groups, according to the Scheuer–Ludwig scoring (S) system.

The inclusion criteria were as follows: (1) clinical diagnosis of chronic hepatitis B; (2) liver biopsy within 3 months; (3) no MRI enhancement contraindications, such as implanted incompatible devices, claustrophobia, and severe renal insufficiency; (4) age >18 years; (5) alcohol consumption <20g/day; and (6) signed informed consent. The exclusion criteria were as follows: (1) combined with other types of viral hepatitis; (2) Child–Pugh score ≥7 points; (3) antiviral, antifibrotic, or antifibrotic treatments before MRI examination; (4) diffuse liver-occupying position; (5) a history of liver surgery or interventional therapy; and (6) difficulty in breathing cooperation. The diagnosis of chronic hepatitis B complied with the diagnostic criteria issued by the European Association for the Study of Liver Diseases in 2017. The baseline characteristics of the patients are shown in Table 1.

Further, 40 of 132 patients were randomly assigned to the test set (with the original distribution of positive and negative cases). The remaining 92 patients constituted the training set for each liver fibrosis classification task (Table 1).

Another 120 patients with hepatocellular carcinoma (HCC) (DCE sequences) having existing manually labeled liver contours were enrolled in this study for transfer learning in liver segmentation.

#### 2.1.2. MRI Protocol

The patient were required to fast for 6 h before the MRI examination. The elimination of interference with image acquisition during the examination was achieved by training for breath-holding and adopting the abdominal band and respiratory gating. A Philips Ingenia 3.0 T MR scanner (Philips Healthcare, Best, the Netherlands) equipped with a dStream Torso Coil body coil was used for scanning. The scan’s field of view included the entire liver, and the contrast agent was injected with Gd-EOB-DTPA at a rate of 2.0 mL/s and a dose of 0.025 mmol/kg. The DCE sequences were acquired within 5 min (mask, arterial, portal venous, and delayed phases) and 20 min (hepatobiliary phase) after contrast agent injection using an mDIXON Water (mDIXON-W) sequence. The mDIXON-W is a 3D T1−weighted gradient echo sequence that applies multiple acquired echo-generating water images. The specific scanning parameters were as follows: flip angle (FA): 10°; echo time (TE1 and TE2): 1.14 and 2.0 ms, respectively; repetition time (TR): 3.3 ms; slice thickness: 3.5 mm; field of view (FOV): 380 × 332 mm^2^; matrix size: 216 × 188; and scanning time: 9.2 s.

#### 2.1.3. Reference Standard

The Scheuer–Ludwig scoring system is the standard for assessing the level of liver fibrosis [41], which encompasses five degrees as follows: S0, not fibrotic; S1, door tube area expansion; S2, fibrosis around the portal area and retention of the leaflet structure; S3, fibrosis with the lobular structural disorder and no cirrhosis; and S4, possible or affirmative cirrhosis. In this study, grades S2–S4 indicated significant fibrosis, whereas grades S3–S4 indicated advanced fibrosis and S4 reflected cirrhosis.

### 2.2. Serum Fibrosis Tests

The aspartate transaminase-to-platelet ratio index (APRI) and the fibrosis-4 index (FIB-4) were recorded, which are the two most widely studied indexes for liver fibrosis assessment, as noninvasive tools [42]. The APRI was calculated as follows: (aspartate aminotransferase (AST) (U/L)/upper limit of the normal AST range × 100)/(platelet count (PLT) (10^9^/L)) and the FIB-4 was calculated as (age (years) × AST (U/L))/(PLT (10^9^/L) × (alanine aminotransferase (ALT) (U/L))^1/2^) [43,44].

### 2.3. Normalized Liver Enhancement

NLE was calculated as the relative enhancement on the pre-contrast images using the following formula: NLE = (SI_HBP_ − SI_DYN1_)/(SI_DYN1_), where SI_DYN1_ and SI_HBP_ are the mean signal intensities of the segmented liver ROI in the mask phase and HBP, respectively. NLE is actually a time-domain feature reported to be associated with the liver fibrosis grade in existing studies [22,23,24,25].

### 2.4. Overall Framework of the Proposed Dynamic Radiomics Model

As illustrated in Figure 1, the framework of the proposed dynamic radiomics model mainly consisted of three steps: (a) liver ROI extraction based on transfer learning, with the post-processing algorithm to exclude interference information; (b) extraction of radiomics features from multi-phase DCE images and through time-domain information; and (c) feature selection, training and evaluation of various classifiers, and prediction in the test set.

### 2.5. Processing Pipeline

#### 2.5.1. ROI Extraction

This study developed a new automatic ROI extraction approach based on 3D liver segmentation and post-processing, which was different from previous manual ROI drawing modes. First, multiple DCE phases were co-registered to the HBP space using a symmetric normalization algorithm [45], performed in Advanced Normalization Tools, which is a state-of-the-art medical image registration and segmentation toolkit. Liver segmentation was implemented on a 3D U-net [46], including multiscale encoding and decoding with convolutional layers, batch normalization layers, max pooling layers, and concatenate structure. The detailed network architecture is shown in Figure 2a. Transfer learning was performed. Specifically, the neural network was pre-trained on a dataset of 120 liver MRI scans with manually labeled liver contours, and fine-tuned and evaluated on the current database (the liver contours of 12 patients in the current database were delineated by an experienced radiologist, including 8 for fine-tuning and 4 for testing). The liver ROIs of the remaining cases were predicted using the trained network.

Moreover, the segmented liver mask was post-processed using image processing techniques, including hole filling, finding the 3D maximum connected component, eroding, and local thresholding [47] to ensure that the extracted ROIs were in the hepatic region and to exclude interference from large portal veins.

#### 2.5.2. Feature Extraction

Feature extraction was first performed separately in each phase of DCE images, collecting radiomics features, including shape and first-order statistical and textural features [48,49]. In total, 14 common shape characteristics described differences in liver shape affected by different degrees of fibrosis, 18 first-order statistical features showed the distribution of voxel intensities, and 73 textural features reflected the internal heterogeneity of the ROI based on five textural matrices. Additionally, some-time domain features were determined for considering the changes in features during the enhancement process. The extracted time-domain features were mainly composed of two parts: (1) the feature discrepancy between different DCE phases and (2) the mean (1), variance (2), skewness (3), kurtosis (4), and entropy (5) of the time-varying curves for each feature except for the common shape features.
(1)$$mean\left(X=[x_{1}\ldots x_{N}\right])=\frac{1}{N}\overset{N}{\underset{i=1}{\sum }}x_{i}$$
(2)$$Var\left(X=[x_{1}\ldots x_{N}\right])=\frac{1}{N-1}\overset{N}{\underset{i=1}{\sum }}{\left(x_{i}-\bar{X}\right)}^{2}$$
(3)$$Skew\left(X=[x_{1}\ldots x_{N}]\right)=\frac{1}{N}\overset{N}{\underset{i=1}{\sum }}{\left[\frac{x_{i}-\bar{X}}{\sigma }\right]}^{3}$$
(4)$$Kurt\left(X=[x_{1}\ldots x_{N}]\right)=\left\{\frac{1}{N}\overset{N}{\underset{i=1}{\sum }}{\left[\frac{x_{i}-\bar{X}}{\sigma }\right]}^{4}\right\}-3$$
(5)$$Entro\left(X=[x_{1}\ldots x_{N}]\right)=-\overset{N}{\underset{i=1}{\sum }}\left(x_{i}^{norm}\log_{2}\left(x_{i}^{norm}+\varepsilon \right)\right)\;x_{i}^{norm}=\frac{\left[x_{i}-\min\left(X\right)\right]}{\left[\max\left(X\right)-\min\left(X\right)\right]}\;\;\;if\;x_{i}\geq 0\;x_{i}^{norm}=\frac{\left[\max\left(X\right)-x_{i}\right]}{\left[\max\left(X\right)-\min\left(X\right)\right]}\;\;\;if\;x_{i}<0\;x_{i}^{norm}=\frac{x_{i}^{norm}}{\sum^{\;}X}$$
where *N* = 5 corresponds to five DCE phases (mask, arterial, portal venous, delayed, and hepatobiliary phases), $\sigma =\sqrt{Var\left(x_{1}\ldots x_{N}\right)}$ is the distribution’s standard deviation, and ε=1e−16 is an extremely small value to ensure the computability of the equation. Finally, a total of 1379 features were included in the feature base, including 471 spatial domain characteristics, 455 feature discrepancy features and 455 time-varying curve-based features. A detailed breakdown of the extracted features is shown in Appendix A. After feature extraction, each feature was normalized using the z-score method based on its distribution in the training set, and the same mean and standard deviation were applied to the normalization of the test set to avoid the overfitting effect introduced by feature normalization.

#### 2.5.3. Feature Selection and Classification

Feature selection aims to obtain the smallest subset of features to avoid the potential problem of overfitting [50] without reducing classification accuracy and retaining the classification distribution. In this study, features were first filtered based on correlation. Specifically, Spearman correlation was performed for correlation analysis, and the permutation test was applied for statistical analysis. Features with correlation coefficients greater than 0.2, and P values less than 0.05, were retained. The least absolute shrinkage and selection operator (LASSO) logistic regression algorithm [51] was further used to filter out the five most important features in the feature subset for classification. Due to the limited sample size, this study adopted a small number of features (top five features) for classifier construction underneath insignificance reduction in training performance to guarantee the generalizability of the model.

In total, six mainstream classifiers, including logistic regression (LR), linear discriminant (LD), k-nearest-neighbor (KNN), Gaussian naive Bayes (GNB), decision tree (DT), and support vector machine (SVM), were trained and evaluated through five-fold cross-validation in the training set. The optimal classifier with the highest AUC value was selected to make predictions in the test set.

### 2.6. Data Resampling

In this study, the data distribution in classification tasks was unbalanced and might lead to biased prediction. The synthetic minority oversampling technique (SMOTE) approach [52] was employed to generate a new balanced data set for subsequent analysis. In specific, the SMOTE algorithm generates new instances from existing minority cases through interpolating new feature values between the target instance and its neighbors in the feature space, while keeping the number of majority cases unchanged. In this study, the number of neighbors to be considered was set as 3. Oversampling was only performed in the training set, and data in the validation and test sets were kept real without interpolation. The resampling process was applied after feature selection because most variable selection algorithms were based on the assumption of sample independence [53].

### 2.7. Evaluation Metrics

The Dice coefficient was used to evaluate the performance in liver segmentation. The performance of the radiomics model was evaluated using the classification accuracy, AUC, average precision (AP), and F1 score. The receiver operating characteristic (ROC) curve, precision-recall (PR) curve, and violin graph of distribution were constructed to demonstrate the performance of the model visually.

### 2.8. Implementation Details

The liver segmentation model based on the 3D U-net architecture was trained and tested with Keras (2.3.1, backend TensorFlow 1.14.0) on a 32 GB NVIDIA TESLA 39C graphics processing unit (GPU). Adaptive moment estimation (Adam) was used as an optimizer in the training procedure, with an initial learning rate of 0.001, and reduced to half of the original value every 100 epochs. The total number of training epochs was set to 500, with a batch size of 5. All volumetric data were downsized to 80 × 256 × 256 and fed into the network in an overlapping manner (eight overlapping layers) with a size of 16 × 256 × 256 during the training process (Figure 2b). In prediction, we only considered the prediction results of the middle eight layers (Figure 2c). The radiomics features were extracted using an open-source ”Pyradiomics“ package (https://pyradiomics.readthedocs.io/en/latest/features.html, version 2.1.2). All classifiers were trained and tested using the ”sklearn“ package (version 0.22.1). The quadratic classification issue (S1–S4) was turned into three binary classification tasks due to the limited number of cases, significant fibrosis (S1 vs. S2–S4), advanced fibrosis (S1–S2 vs. S3–S4) and cirrhosis (S1–S3 vs. S4) classification tasks. Statistical evaluation was based on the subject level in this study, implying that the classifications were performed on each patient rather than on each slice. The Mann–Whitney test was performed to compare the selected feature values and predicted scores between the positive and control groups. A *p*-value < 0.05 indicated a statistically significant difference.

## 3. Results

### 3.1. Liver Segmentation

Liver segmentation based on deep learning achieved a 3D volume Dice score of 0.961 ± 0.007 in the test set of current datasets. A segmentation slice demonstration is illustrated in Figure 3. The segmented mask fitted the real label well, and the post-processing algorithms effectively removed interfering information such as large vessels.

### 3.2. Proposed Dynamic Radiomics Model

A total of five features were included in the final feature subset in each classification task. The corresponding regularization parameters in LASSO logistic regression was 0.103, 0.147, and 0.129 for staging significant fibrosis, advanced fibrosis, and cirrhosis, respectively. The selected features and their importance are demonstrated in Figure 4 (left panel). In the classification of significant fibrosis and advanced fibrosis, three of the selected features belonged to the variation of textural features in the time domain, one was the variation of histogram variable in the time domain, and the other was textural feature in hepatobiliary phase. In cirrhosis classification, two important variables were shape-based features, two were the variation of textural features in the time domain, and the other belonged to textural feature in delayed phase. The Mann–Whitney test was performed for each selected feature, and *p*-values were further corrected by false discovery rate (FDR) method. The results are shown in Table 2.

It can be seen that after correction, one feature in significant fibrosis classification and four features in advanced fibrosis and cirrhosis classifications are statistically significant. In addition, the values of features in two subsets of patients with extreme classifier prediction scores were investigated, one was chosen as the 20% with the lowest prediction scores in the control group, and the other was 20% with the highest prediction scores in the positive group [54]. The results are shown in Table 3.

According to results of statistical test for each feature and statistics of feature values for patients with extreme values of the prediction scores by the classifier, it can be found that there was an obvious difference in studied features between the positive and control groups, and this difference was more significant between two subgroups with extreme classifier prediction scores. In addition, the way features change in the time domain undoubtedly played an important role in the liver fibrosis grading model, which accounted for a large part of the selected features. For example, dependence variance measured the variance in dependence size in the liver region based on gray level dependence matrix (GLDM), and it showed a bigger difference between arterial phase and mask phase for significant fibrosis cases. Moreover, informational measure of correlation assessed the correlation between the probability distributions of different grey levels based on gray level co-occurrence matrix (GLCM), which can quantify the complexity of the texture, and both in advanced fibrosis and cirrhosis, a smaller difference was obtained between HBP and earlier phase for higher grade liver fibrosis. Furthermore, the morphology of the liver changed significantly in cirrhosis stage, with a larger diameter in sagittal plane, and a smaller sphericity value, which was important for identifying patients with cirrhosis. The image comparison between a typical cirrhotic patient and a non-cirrhotic patient is shown in Figure 5, and the normalized feature values for these two cases were as follows: GLDM Large Dependence High Gray Level Emphasis in DYN4 (positive: −1.174, control: 1.395); Shape-based Maximum 2D diameter (positive: 0.370, control: −0.793); Shape-based Sphericity (positive: −0.563, control: 0.032); GLCM Informational Measure of Correlation (HBP-DYN4) (positive: −2.273, control: 1.493); and GLCM Kurtosis of Cluster Prominence in time domain (positive: 1.259, control: −1.285). Liver tumors were common in patients with cirrhosis, it can be found from Figure 5a that the liver portion used for feature extraction did not include the tumor region and thus the interference information was well excluded, which reflected the effectiveness of the post-processing algorithms.

The performance of the six classifiers in five-fold cross-validation and the corresponding AUC value in the test set of the selected optimal classifier are shown in Figure 6 (top panel). For significant fibrosis, the best performance classifier was GNB with an AUC value of 0.874 ± 0.078 in the validation set. The selected classifier achieved an accuracy of 0.875, with an AUC value of 0.867 (95% confidence interval (CI): 0.723–0.954), an AP score of 0.939, and an F1 score of 0.921 in the test set. In the stage of advanced fibrosis, SVM was selected with a validation AUC value of 0.883 ± 0.080, whereas the performance in the test set had an accuracy of 0.825, an AUC value of 0.874 (95%CI: 0.730–0.957), an AP score of 0.906, and an F1 score of 0.837. The selected top classifier was LR with an AUC value of 0.897 ± 0.059 in cross-validation, and achieved an accuracy of 0.850, with an AUC value of 0.900 (95%CI: 0.764–0.972), an AP score of 0.866, and an F1 score of 0.750 in the test set in cirrhosis cases. The prediction scores generated by the classifier for each case in the test set are shown in Figure 4 (right panel), the scores (median (interquartile range)) of positive patients were significantly higher than the control group in significant fibrosis classification: 0.971 (0.942~0.983) vs. 0.210 (0.050~0.719), *p* < 0.001; advanced fibrosis classification: 0.764 (0.552~0.837) vs. 0.283 (0.170~0.392), *p* < 0.001; and cirrhosis classification: 0.768 (0.303~0.897) vs. 0.049 (0.018~0.208), *p* < 0.001. The confusion matrix in the test set for each classification task is illustrated in Figure 6 (bottom panel).

### 3.3. Performance Comparison

The performance of the proposed model for fibrosis grading was compared with those of single-phase DCE-based radiomics model (radiomics features from the mask phase, arterial phase, portal venous phase, delayed phase, and HBP, individually), multi-phase DCE-based radiomics model (combination of imaging features from the mask phase, arterial phase, portal venous phase, delayed phase, and HBP, without time-domain features), NLE, and serological indicators including APRI and FIB-4. The number of selected features in the single-phase or multi-phase DCE-based radiomics models was five; the training and test sets remained unchanged, and the entire radiomics analysis followed the same process with the proposed dynamic radiomics model.

The ROC and PR curves of various liver fibrosis staging models are shown in Figure 7. The dynamic radiomics model had the most stable and best overall performance in each classification task, with AUC improvements of at least 0.03 in significant fibrosis, 0.05 in advanced fibrosis, and 0.01 in cirrhosis classification. Moreover, the prediction accuracy of the dynamic radiomics model was also the highest in each classification task. Table 4 illustrates the specific performance of each model in terms of accuracy, AUC, AP score, and F1 score. Furthermore, violin graphs were constructed for various models to observe their discrimination abilities in liver fibrosis grading. As shown in Figure 8, the proposed radiomics model had the best discriminatory potential for each classification task.

## 4. Discussion

This study aimed to assess hepatic fibrosis based on dynamic radiomics analysis combining multiple DCE phases and time-domain information. The proposed method demonstrated a considerable value in classifying significant fibrosis, advanced fibrosis, and cirrhosis, indicating that the dynamic radiomics analysis of Gd-EOB-DTPA MRI images might have the potential for automatic hepatic fibrosis staging in patients with hepatitis B. However, physicians need to depend on liver biopsy tests for grading in general.

Non-invasive hepatic fibrosis assessment is a hot topic and has been explored in multiple studies. One of the main directions is to measure liver stiffness, usually by ultrasound elastography [29,30] or MRE [33,34,35]. This approach shows a high correlation between the degree of fibrosis and liver stiffness. However, ultrasound-based measurement methods may suffer from unreliable issues due to the lack of reproducibility and operator dependence [31]. MRE can ameliorate these drawbacks to some extent, but requires additional equipment and is more expensive compared with conventional MRI. Therefore, radiomics analysis has attracted increasing attention recently. The discriminating ability of radiomics analysis has a strong correlation with histopathological characteristics. The destruction of liver parenchyma homogeneity is reflected in the liver by textural features [17] and characterized by fibrous septa and nodules of different sizes because lymphocyte infiltration and hepatocyte damage are the characteristics of the necroinflammatory process in chronic viral hepatitis [55,56]. Most established radiomics studies have been performed on CT and MRI images. According to a study by Zhang et al. [13], MRI images generally have an advantage over CT images in hepatic fibrosis evaluation based on radiomics analysis. In another study, Watanabe et al. [57] pointed to a conspicuous correlation between the contrast enhancement index in gadoxetate disodium-enhanced MRI and fibrosis staging. Thus, enhanced MRI is more reliable for staging hepatic fibrosis than diffusion-weighted MRI, besides hematologic and clinical parameters. Furthermore, previous studies demonstrated the potential of liver-specific Gd-EOB-DTPA-enhanced MRI in hepatic fibrosis assessment, in which the HBP after 20 min highly correlated with the degree of liver fibrosis [22,23,24,25,26,27]. In this study, we performed a systematic work on radiomics analysis based on Gd-EOB-DTPA-enhanced MRI. A dynamic radiomics model was proposed, which used radiomics features from multi-phase DCE images as well as time-domain features through the enhancement process. Time-domain information played an important role in liver fibrosis staging, which was not discussed previously.

This study was novel in proposing fully automatic dynamic radiomics analysis for hepatic fibrosis staging based on Gd-EOB-DTPA-enhanced MRI. The proposed model had the most stable and best overall performance in all stages of classification compared with the staging results of conventional single-phase or multi-phase DCE-based radiomics models, NLE, and serological indicators. Meanwhile, this study found that the HBP was also critical to hepatic fibrosis staging, outperforming other individual phases in classification, corroborating previous findings. Moreover, a simple combination of imaging features from five DCE phases did not achieve better results and was even worse than using HBP features alone. Indeed, the time-domain features played a key role in the dynamic radiomics model.

The strengths of this study were as follows. First, in previous studies, most texture analyses of hepatic fibrosis were performed on a manual extraction area at a representative slice, for example, a circular or square region [12,13,14,16,17]. Alternatively, the entire liver delineated on the slice contained the largest cross-section through the liver, carefully avoiding the portal vein, very large intrahepatic vessels, and any obvious motion-affected regions [15]. This pattern of ROI extraction was extremely time consuming, and might result in the loss or destruction of valid texture information due to the investigator’s experience and other uncontrollable factors. The choice of ROI might also heavily influence the results of radiomics models. Different from previously used methods, an automatic liver ROI extraction strategy based on transfer learning was applied in this study and the segmented mask was post-processed to remove interfering information effectively. This fully automatic extraction method could reduce the workload of the radiologist to a large extent, and eliminate the influence of region selection on the results of radiomics analysis. Second, the radiomics features from multiple DCE phases and the time-domain features extracted from the time-varying curves and characteristic differences in imaging features through the enhancement process were combined in the feature extraction step to establish a larger feature base. Hence, a more valuable feature combination was explored, resulting in superior classification performance. Although only five features were covered after feature selection in the classification model, the time-domain features accounted for a large proportion of the final feature subset, indicating the critical role of time-domain features in the grading of liver fibrosis. Specifically, 80% of the features ultimately contributing to the classification model were time-domain features in significant fibrosis and advanced fibrosis classification, while 40% were time-domain features in cirrhosis classification. In addition, feature extraction was performed based on the segmented liver volume, hence some valuable morphological features of the liver could be considered. As a result, two shape-based features were selected in cirrhosis classification, revealing significant changes in liver morphology in this stage. This was also supported by a previous study [58], reporting that the shape of the liver could be altered to some extent due to hepatocyte necrosis and collapse. The exploitation of shape features was also not possible with the traditional circular or square ROI extraction mode. Considering the aforementioned improvements, the proposed model had the most stable and best overall performance in hepatic fibrosis staging tasks, while being more automated.

Although the proposed dynamic radiomics model has achieved superior performance on the current dataset, the generalization of the model is still a challenging task. MRI-based radiomics features can be affected by many factors, for example, multiple MR scanners, MR acquisition parameters, reconstruction algorithms, introduced noise and artefacts, and pre-processing and post-processing of images or features [59,60,61,62]. We believe that the first priority is to standardize the entire process, that is, for future data collection, the acquisition parameters and processing algorithms should remain as consistent as possible with the developed standards. In addition, the assessment of liver fibrosis was based on the whole liver region in this study, so it can be inferred that the model was robust to local noise and artifacts. The biggest concern is the inevitable data fluctuations that occur when models are applied to multi-center data, due to differences in MR scanners and scanning protocols. Fortunately, the acquisition process of Gd-EOB-DTPA-enhanced MRI is relatively fixed in different centers, with similar timing of acquisition for each enhanced phase after contrast injection (arterial phase, portal venous phase, delayed phase, and hepatobiliary phase). For the differences in image intensity distribution in multi-center datasets, two strategies can be explored, (1) for radiomics model, the reproducibility and stability of radiomics features can be evaluated and the features that are more robust across multi-center data can be selected to build classification models [63,64]. (2) Some intensity standardization algorithms can be applied to the preprocessing of multi-center image data to adjust the intensity distribution, for example, global histogram matching algorithm, joint histogram registration algorithm, and generative adversarial network-based method [65].

This study had some limitations. The sample size needed to be enlarged, and the generalizability of the hepatic fibrosis model needed further validation in a multi-center large-scale study. In addition, this study was conducted for Gd-EOB-DTPA-enhanced MRI, and the effect of dynamic information in conventional MRI contrast agents needed to be further tested. Finally, the value of the model in grading non-diffuse hepatic fibrosis remains unknown. In future work, a multi-center large-scale data set will be collected prospectively, and the classification performance and generalization ability of the proposed dynamic radiomics model are expected to be further improved.

## 5. Conclusions

This study proposed an automatic hepatic fibrosis grading model based on dynamic radiomics analysis in multiple Gd-EOB-DTPA-enhanced DCE phases. The whole process was highly automated, which could save time and energy for new case prediction or additional dataset training. The prediction performance of the dynamic radiomics model was more superior for the classification of all stages of hepatic fibrosis compared with the conventional single-phase or multi-phase DCE-based radiomics models, NLE, and some clinical serum parameters, indicating an association of fibrosis stage with dynamic radiomics features.

## Figures and Tables

**Figure 1 biomolecules-11-00307-f001:**
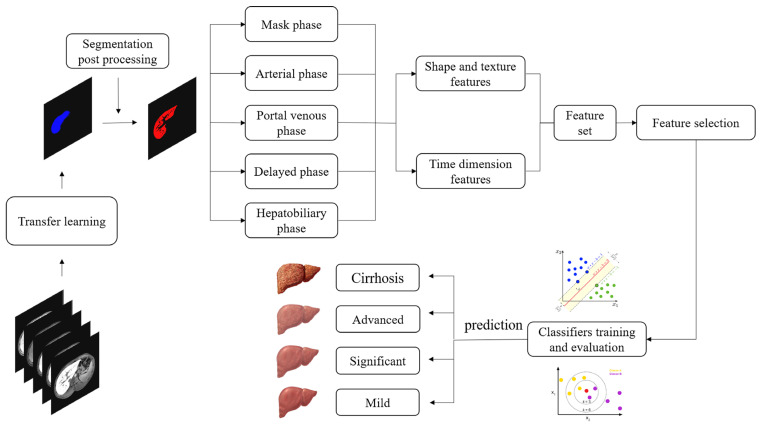
Overall framework of the proposed dynamic radiomics model, including liver ROI extraction, features extraction, feature selection, and classification.

**Figure 2 biomolecules-11-00307-f002:**
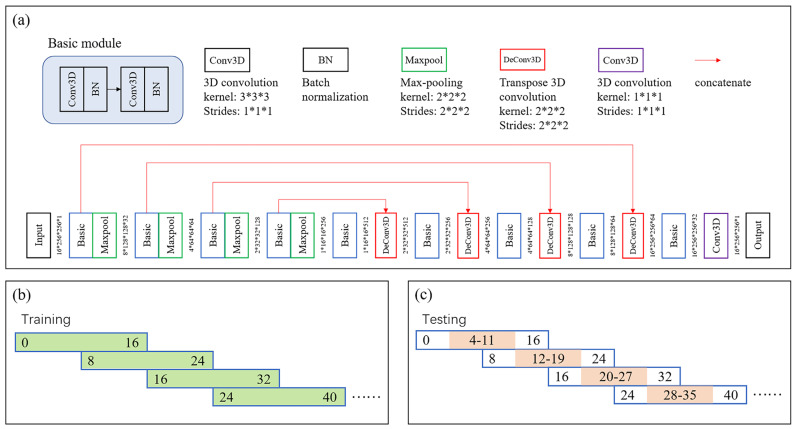
Three-dimensional U-net in liver segmentation. (**a**) Network architecture; (**b**) training strategy: image patches with eight slices of overlap; and (**c**) testing strategy: image patches with eight slices of overlap and retention of prediction results of middle eight slices.

**Figure 3 biomolecules-11-00307-f003:**
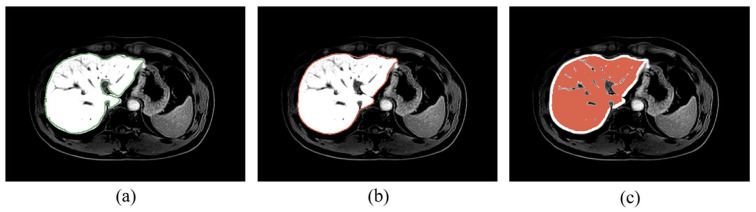
Liver segmentation results. (**a**) Manually labeled liver contour; (**b**) predicted liver contour through deep learning; and (**c**) final segmentation mask after post-processing.

**Figure 4 biomolecules-11-00307-f004:**
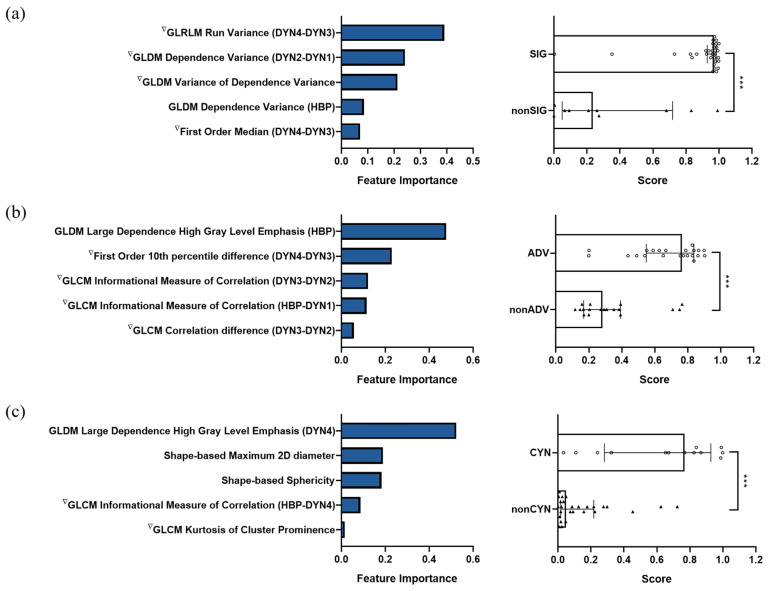
Selected features with corresponding feature importance (left panel) and prediction scores in positive and control groups (right panel) for (**a**) significant fibrosis, (**b**) advanced fibrosis, and (**c**) cirrhosis classification. GLCM, gray level co-occurrence matrix; GLDM, gray level dependence matrix; GLRLM, gray level run length matrix. Significance level: *** *p* < 0.001.

**Figure 5 biomolecules-11-00307-f005:**
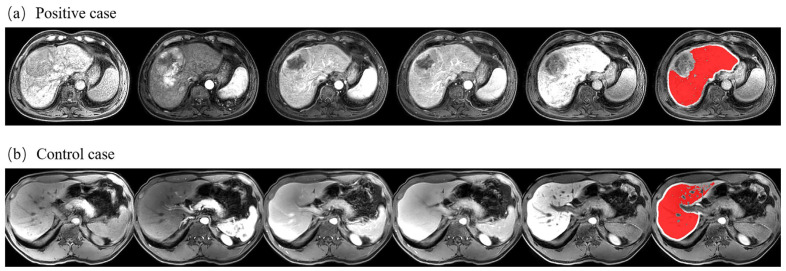
Comparison of cirrhotic and non-cirrhotic images. (**a**) One patient from the positive group (S = 4); and (**b**) one patient from the control group (S = 1). From left to right: DCE images from the pre-contrast phase, arterial phase, portal venous phase, delayed phase, and hepatobiliary phase. The rightmost column represents the segmented liver mask after post-processing.

**Figure 6 biomolecules-11-00307-f006:**
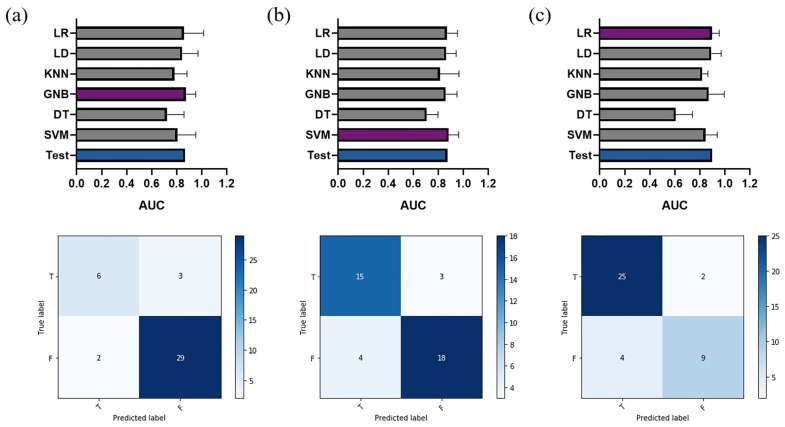
Performances of various classifiers in five-fold cross-validation and the corresponding AUC value in the test set of the selected optimal classifier (top panel); confusion matrix in the test set (bottom panel) for (**a**) significant fibrosis, (**b**) advanced fibrosis, and (**c**) cirrhosis classification. Purple bar: optimal classifier in cross-validation; blue bar: corresponding AUC value in the test set.

**Figure 7 biomolecules-11-00307-f007:**
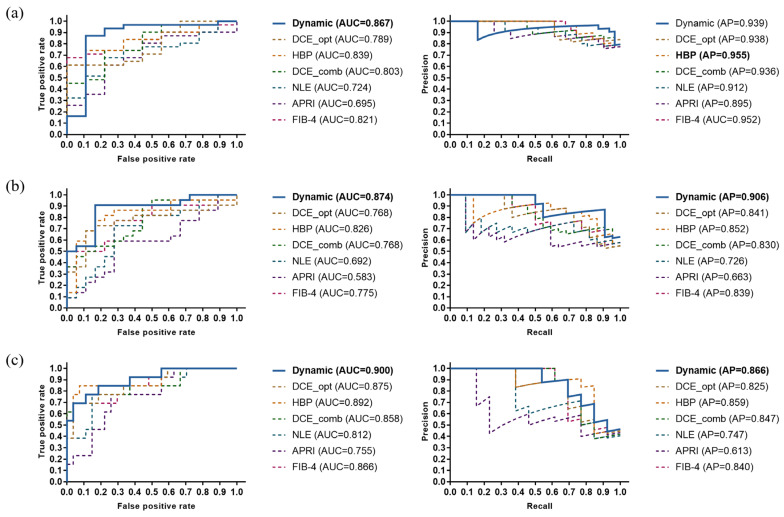
ROC curves (left panel) and PR curves (right panel) in the test set for various models in (**a**) significant fibrosis, (**b**) advanced fibrosis, and (**c**) cirrhosis staging. Dynamic, dynamic radiomics model; DCE_opt, single-phase DCE-based radiomics model with best overall performance between the mask phase, arterial phase, portal venous phase, and delayed phase (delayed phase for significant fibrosis, mask phase for advanced fibrosis, and arterial phase for cirrhosis); HBP, hepatobiliary phase based-radiomics model; DCE_comb, multi-phase DCE-based radiomics model; NLE, normalized liver enhancement model; APRI, aspartate transaminase-to-platelet ratio index model; FIB-4, fibrosis-4 index model.

**Figure 8 biomolecules-11-00307-f008:**
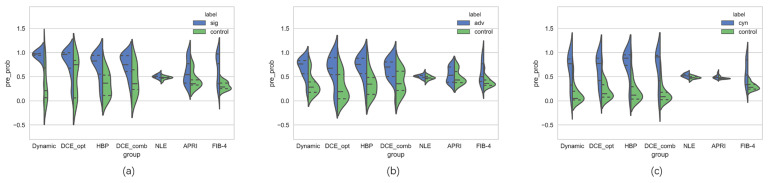
Violin graph of the distribution of various models in the test set for (**a**) significant fibrosis, (**b**) advanced fibrosis, and (**c**) cirrhosis staging. Dynamic, dynamic radiomics model; DCE_opt, single-phase DCE-based radiomics model with best overall performance between the mask phase, arterial phase, portal venous phase and delayed phase (delayed phase for significant fibrosis, mask phase for advanced fibrosis, and arterial phase for cirrhosis). HBP, hepatobiliary phase-based radiomics model; DCE_comb, multi-phase DCE-based radiomics model; NLE, normalized liver enhancement model; APRI, aspartate transaminase-to-platelet ratio index model; FIB-4, fibrosis-4 index model. The dotted lines in the graph indicate the median and interquartile range.

**Table 1 biomolecules-11-00307-t001:** Baseline characteristics of the study population and patient statistics.

	Overall	Significant		Advanced		Cirrhosis	
Sex							
Male	93 (70.5%)	79 (77.5%)		61 (82.4%)		36 (85.7%)	
Female	39 (29.6%)	23 (22.6%)		13 (17.6%)		6 (14.3%)	
Age (years) *	45.8±13.2	47.7 ± 13.3		50.8 ± 12.6		52.6 ± 11.1	
Fibrosis score							
S1	30 (22.7%)	0 (0%)		0 (0%)		0 (0%)	
S2	28 (21.2%)	28 (27.45%)		0 (0%)		0 (0%)	
S3	32 (24.2%)	32 (31.4%)		32 (43.2%)		0 (0%)	
S4	42 (31.8%)	42 (41.2%)		42 (56.8%)		42 (100%)	
Group		+	–	+	–	+	–
Training		71	21	52	40	29	63
Test		31	9	22	18	13	27

* Data are mean ± standard deviation; + represents the positive patient and – represents the control group; S means the Scheuer–Ludwig degree.

**Table 2 biomolecules-11-00307-t002:** Comparison of selected features between positive and control groups and the results of statistical tests.

Feature	Positive	Control	*p*-Value	Adjusted *p*-Value
**Significant fibrosis**				
^∇^GLRLM, Run Variance (DYN4-DYN3)	−0.518 (−0.675~−0.257)	0.181 (−0.643~0.934)	0.067	0.084
^∇^GLDM, Dependence Variance (DYN2-DYN1)	0.208 (−0.088~0.557)	−0.442 (−1.275~0.117)	0.006 ^**^	0.028 ^*^
^∇^GLDM, the variance of Dependence Variance in time domain	−0.617 (−0.716~−0.110)	−0.247 (−0.410~0.495)	0.037 ^*^	0.077
GLDM, Dependence Variance in HBP	−0.479 (−0.953~0.198)	0.717 (−0.521~1.271)	0.046 ^*^	0.077
^∇^First Order, Median (DYN4-DYN3)	0.169 (−0.837~0.868)	0.527 (−0.192~1.202)	0.308	0.308
**Advanced fibrosis**				
GLDM, Large Dependence High Gray Level Emphasis in HBP	−0.310 (−0.920~0.005)	0.288 (−0.451~0.502)	0.030 ^*^	0.037 ^*^
^∇^First Order, 10th percentile difference (DYN4-DYN3)	0.048 (−0.548~0.574)	0.204 (−0.224~1.076)	0.265	0.265
^∇^GLCM Informational Measure of Correlation (DYN3-DYN2)	0.914 (0.214~1.412)	−0.258 (−0.733~0.416)	0.002 ^**^	0.006 ^**^
^∇^GLCM Informational Measure of Correlation (HBP-DYN1)	−0.645 (−0.864~−0.048)	0.648 (−0.279~1.357)	0.004 ^**^	0.006 ^**^
^∇^GLCM Correlation difference (DYN3-DYN2)	−0.680 (−0.961~0.039)	0.400 (0.038~1.064)	0.002 ^**^	0.006 ^**^
**Cirrhosis**				
GLDM Large Dependence High Gray Level Emphasis in DYN4	−0.784 (−1.066~0.256)	0.601 (−0.403~1.027)	0.010^**^	0.034^*^
Shape-based Maximum 2D diameter	0.429 (−0.060~0.800)	−0.175 (−0.763~0.481)	0.022^*^	0.034^*^
Shape-based Sphericity	−0.699 (−1.151~0.226)	0.366 (−0.168~0.817)	0.027^*^	0.034^*^
^∇^GLCM Informational Measure of Correlation (HBP-DYN4)	−0.428 (−1.392~0.069)	0.212 (−0.255~1.274)	0.020^*^	0.034^*^
^∇^GLCM Kurtosis of Cluster Prominence in time domain	0.457 (0.015~1.162)	−0.725 (−1.140~0.344)	0.055	0.055

^∇^ Time domain features. (i-j), feature difference between i and j. Single feature values (F_i_) were normalized (F_i,norm_) with total feature means and standard deviations as F_i,norm_ = (F_i_-mean (F_i_))/SD (F_i_). Feature values are listed as median (Interquartile range). *p*-values were calculated based on Mann–Whitney test and the adjusted *p*-values were corrected by false discovery rate method. DYN1, mask phase; DYN2, arterial phase; DYN3, portal venous phase; DYN4, delayed phase; HBP, hepatobiliary phase; GLCM, gray level co-occurrence matrix; GLDM, gray level dependence matrix; GLRLM, gray level run length matrix. Significance level: * *p* < 0.05, ** *p* < 0.01.

**Table 3 biomolecules-11-00307-t003:** Comparison of selected features between two subsets of patients with extreme classifier prediction scores.

Feature	Positive Subset (20%)	Control Subset (20%)
**Significant fibrosis**		
^∇^GLRLM, Run Variance (DYN4-DYN3)	−0.786	0.289
^∇^GLDM, Dependence Variance (DYN2-DYN1)	0.090	−1.684
^∇^GLDM, the variance of Dependence Variance in time domain	−0.455	0.044
GLDM, Dependence Variance in HBP	−0.775	0.397
^∇^First Order, Median (DYN4-DYN3)	−1.869	0.832
**Advanced fibrosis**		
GLDM, Large Dependence High Gray Level Emphasis in HBP	−0.883	0.149
^∇^First Order, 10th percentile difference (DYN4-DYN3)	−0.263	0.581
^∇^GLCM Informational Measure of Correlation (DYN3-DYN2)	1.176	−0.948
^∇^GLCM Informational Measure of Correlation (HBP-DYN1)	−1.070	1.087
^∇^GLCM Correlation difference (DYN3-DYN2)	−0.984	0.657
**Cirrhosis**		
GLDM Large Dependence High Gray Level Emphasis in DYN4	−0.851	1.050
Shape-based Maximum 2D diameter	0.855	−0.839
Shape-based Sphericity	−1.809	0.531
^∇^GLCM Informational Measure of Correlation (HBP-DYN4)	−1.691	0.901
^∇^GLCM Kurtosis of Cluster Prominence in time domain	1.161	−0.562

^∇^ Time domain features. (i-j), feature difference between i and j. Single feature values (F_i_) were normalized (F_i,norm_) with total feature means and standard deviations as F_i,norm_ = (F_i_-mean (F_i_))/SD (F_i_). Feature values are listed as mean value. DYN1, mask phase; DYN2, arterial phase; DYN3, portal venous phase; DYN4, delayed phase; HBP, hepatobiliary phase; GLCM, gray level co-occurrence matrix; GLDM, gray level dependence matrix; GLRLM, gray level run length matrix. Those were defined as the 20% patients in the positive group with the highest prediction scores and the 20% patients in the control group with the lowest prediction scores.

**Table 4 biomolecules-11-00307-t004:** Detailed performance comparison for the proposed model, single-phase or multi-phase DCE-based radiomics models, NLE, and some clinical serum parameters.

Model	Accuracy	AUC (95%CI)	AP	F1
**Significant fibrosis**				
Dynamic (proposed)	0.875	0.867 (0.723~0.954)	0.939	0.921
DYN1	0.750	0.778 (0.619~0.894)	0.934	0.839
DYN2	0.575	0.581 (0.414~0.735)	0.855	0.667
DYN3	0.725	0.814 (0.659~0.919)	0.922	0.792
DYN4	0.775	0.789 (0.631~0.901)	0.938	0.857
HBP	0.750	0.839 (0.688~0.936)	0.955	0.828
DYN combined	0.775	0.803 (0.647~0.911)	0.936	0.852
NLE	0.625	0.724 (0.560~0.853)	0.912	0.706
APRI	0.625	0.695 (0.530~0.831)	0.895	0.706
FIB-4	0.725	0.821 (0.667~0.924)	0.952	0.784
**Advanced fibrosis**				
Dynamic (proposed)	0.825	0.874 (0.730~0.957)	0.906	0.837
DYN1	0.750	0.768 (0.607~0.886)	0.841	0.773
DYN2	0.525	0.710 (0.545~0.842)	0.768	0.387
DYN3	0.575	0.692 (0.526~0.828)	0.716	0.485
DYN4	0.650	0.715 (0.550~0.846)	0.754	0.667
HBP	0.800	0.826 (0.673~0.927)	0.852	0.818
DYN combined	0.675	0.768 (0.607~0.886)	0.830	0.723
NLE	0.700	0.692 (0.526~0.828)	0.726	0.714
APRI	0.575	0.583 (0.417~0.737)	0.663	0.541
FIB-4	0.675	0.775 (0.616~0.892)	0.839	0.629
**Cirrhosis**				
Dynamic (proposed)	0.850	0.900 (0.764~0.972)	0.866	0.750
DYN1	0.775	0.852 (0.704~0.944)	0.825	0.640
DYN2	0.850	0.875 (0.732~0.958)	0.825	0.750
DYN3	0.825	0.855 (0.707~0.946)	0.833	0.741
DYN4	0.775	0.832 (0.680~0.931)	0.785	0.640
HBP	0.850	0.892 (0.753~0.968)	0.859	0.786
DYN combined	0.850	0.858 (0.711~0.948)	0.847	0.750
NLE	0.725	0.812 (0.657~0.918)	0.747	0.645
APRI	0.725	0.755 (0.593~0.877)	0.613	0.522
FIB-4	0.850	0.866 (0.721~0.953)	0.840	0.700

Abbreviations: AUC, Area under the curve; 95%CI, 95% confidence interval; AP, average precision; Dynamic, dynamic radiomics model; DYN1, mask phase-based radiomics model; DYN2, arterial phase-based radiomics model; DYN3, portal venous phase-based radiomics model; DYN4, delayed phase-based radiomics model; HBP, hepatobiliary phase-based radiomics model; DYN combined, multi-phase DCE-based radiomics model; NLE, normalized liver enhancement model; APRI, aspartate transaminase-to-platelet ratio index model; FIB-4, fibrosis-4 index model.

## Data Availability

The example data and extracted features that support the findings of this study is openly available at link: http://180.167.250.222:8888/d/f78490a9288d4c6dbdeb/.

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
