# Peer review of "Imaging-Based Staging of Hepatic Fibrosis in Patients with Hepatitis B: A Dynamic Radiomics Model Based on Gd-EOB-DTPA-Enhanced MRI"

_biomolecules, 2021, doi:10.3390/biom11020307_

Round 1

Reviewer 1 Report

Journal: Biomolecules

Manuscript Title: Imaging-based staging of hepatic fibrosis in patients with hepatitis B: a dynamic radiomics model based on Gd-EOB-DTPA-4 enhanced MRI

Authors: R Zheng, C Shi, C Wang, N Shi, T Qiu, W Chen, Y Shi, and H Wang

Major concerns to the approach and study design include:

1) This work has a range of limitations, chief among which is that the study is retrospective and from a single institute, further coupled with a very limited number of patients, as has been acknowledged by the authors. As such, the proposed radiomics model may have limited utility in broader clinical practices. Prospective study should be considered, and such study would be very feasible and most likely leads to much more viable and clinically relevant results.

2) It is well known that radiomics parameters are susceptible to intra-/inter-scanner sensitivity fluctuation (PMID: 14972397) and typically the raw intensity data undergoes a normalization process in order to alleviate this effect. The authors are advised to take this into consideration.

3) The statistics analysis that the authors took to examine the discriminative power of the extracted radiomics features in classifying liver fibrosis stages is problematic. Given the analysis performed in a setting where multiple comparisons arise, p-values for significance should be adjusted accordingly, which may, in turn, imperil the viability of some of the claimed findings.

4) A number of previous studies have shown the impacts on MR radiomics analysis due to variability of acquisition parameters and postprocess variables (PMID:29891091; PMID: 32277703; PMID:30154684; PMID: 32277703). To explore how these affect the proposed radiomics model in detail may be out of the scope of the current study, but discussions are definitely merited, such as how and in what way the proposed model behaves in response to those stochastic effects.

5) Caution should be exerted when using SMOTE to oversample under-represented class since it blindly generalizes the minority without regard to the majority class and this is particularly problematic in the cases of highly skewed class distributions as is typically seen in radiomics analysis.

6) The paper has typos, and the writing style is not professional enough. The manuscript could be further improved by asking senior professionals to revise language for style and grammar problems.

Reviewer 2 Report

This is a very well conducted and written radiomics work about a model for predicting hepatic fibrosis using radiomics. The methods are descripted with much details.

I have only minor comments, mostly regarding providing interpretation of features selected: 

  • can the authors perform tests to see if the features selected in the model are correlated with fibrosis? auhtors only show AUC of features, while a statistical test to assess if features are different between patients with fibrosis and patients without would be interesting.
  • of all the machine learning methods used, why GNB performed best?
  • The authors should interpretate the radiomics features found as significant, which properties of liver they describe (e.g. inhomogeneity, intensity) and how they are (higher/lower) on patients with high risk/low risk for fibrosis. A possible approach is to study feature values on patients with high/low risk, defined as those with lowest/highest score function, as described in Avanzo et al , Front. Oncol., 21 April 2020 | https://doi.org/10.3389/fonc.2020.00490, https://www.frontiersin.org/articles/10.3389/fonc.2020.00490/full  which is recommended citing. Also is recommended showing figures of low risk and high risk patients and discussing their appearance and features values.
  • two shape features were selected, what do they mean about the shape of the high risk/low risk patient?
  • briefly describe in the methods which features were extracted and their definitions (briefly cite some papers with definitions, e.g. Image Biomarker Standardization Initiative) 

Round 2

Reviewer 1 Report

The revised manuscript is much improved, and I am pleased with the overall responses by the authors. I believe that this work, as presented now, will be a welcome contribution to the hepatic fibrosis research despite a number of limitations in the study as the authors acknowledged. I have no further major comments.

Reviewer 2 Report

The authors responded to all my concerns and the manuscript can be accepted.